

# The relationship between sustained attention and aerobic fitness in a group of young adults

Luis F. Ciria[1,2], Pandelis Perakakis[1], Antonio Luque-Casado[1,2], Cristina Morato[1] and Daniel Sanabria[1,2]

[1] Mind, Brain & Behavior Research Center, University of Granada, Granada, Spain
[2] Departamento de Psicología Experimental, University of Granada, Granada, Spain

## ABSTRACT

**Background**. A growing set of studies has shown a positive relationship between aerobic fitness and a broad array of cognitive functions. However, few studies have focused on sustained attention, which has been considered a fundamental cognitive process that underlies most everyday activities. The purpose of this study was to investigate the role of aerobic fitness as a key factor in sustained attention capacities in young adults.

**Methods**. Forty-four young adults (18–23 years) were divided into two groups as a function of the level of aerobic fitness (high-fit and low-fit). Participants completed the Psychomotor Vigilance Task (PVT) and an oddball task where they had to detect infrequent targets presented among frequent non-targets.

**Results**. The analysis of variance (ANOVA) showed faster responses for the high-fit group than for the low-fit group in the PVT, replicating previous accounts. In the oddball task, the high-fit group maintained their accuracy (ACC) rate of target detection over time, while the low-fit group suffered a significant decline of response ACC throughout the task.

**Discussion**. Importantly, the results show that the greater sustained attention capacity of high-fit young adults is not specific to a reaction time (RT) sustained attention task like the PVT, but it is also evident in an ACC oddball task. In sum, the present findings point to the important role of aerobic fitness on sustained attention capacities in young adults.

## INTRODUCTION

Sustained attention is a fundamental high-order function that enables to direct and focus cognitive activity on specific stimuli over prolonged periods of time, and is required in order to complete any planned activity, sequenced action, or thought. Sustained attention is therefore the basis for many types of information processing and it is crucial for healthy cognitive development that allows both, proper adaptation to environmental demands, and the capacity to modify behavior (*Sarter, Givens & Bruno, 2001*). Importantly, sustaining attention for long periods of time is necessary in fundamental everyday activities, such as

Corresponding author
Luis F. Ciria, lciria@ugr.es, lciriaperez@gmail.com

attending to academic lessons at school or driving, and in highly responsible professional tasks like performing surgery or piloting (*Di Stasi et al., 2015*; *Suess, Porges & Plude, 1994*).

It is well known that the capacity to stay focused on a particular task is limited, which results in performance decrements with increasing time-on-task. This effect is known as the vigilance decrement and is thought to reflect the depletion of attentional resources over time (*Helton & Warm, 2008*; *Davies & Parasuraman, 1982*). Therefore, a profound understanding of the factors influencing sustained attention capacities is highly relevant to both theoretical and applied research.

In the present study, we focus on aerobic fitness as a key feature in young adults related to their ability to sustain attention. Although a substantial amount of studies have investigated the effects of aerobic fitness on cognitive performance in general (*Colcombe & Kramer, 2003*; *Hillman, Erickson & Kramer, 2008*), the relation between fitness and sustained attention has not received proper attention by exercise and cognition researchers. Here, we addressed this issue by comparing behavioral performance of high-fit and low-fit young adults in two sustained attention tasks.

In general, the limited previous evidence has indicated that aerobic fitness and sustained attention are positively related across the life span (*Bunce, 2001*; *Prakash et al., 2011*). For instance, in a recent study, Getzmann and collaborators (*2013*) found a higher susceptibility to distraction over time for inactive seniors than their active counterparts. Similar outcomes have been seen in children (*Chaddock et al., 2012*; *Pontifex et al., 2012*). With respect to young adults, to the best of our knowledge, there are only three published studies (*Luque-Casado et al., 2013*; *Luque-Casado et al., 2016a*; *Luque-Casado et al., 2016b*) that have reported significant differences in sustained attention capacities as a function of aerobic fitness (see *Bunce, Barrowclough & Morris, 1996*, for a null result). Luque-Casado and collaborators (*2013*) reported the first direct evidence of a selective association between aerobic fitness and sustained attention in young adults. They found overall faster responses in a short (i.e., 10 min) Psychomotor vigilance task (PVT) in high fit as compared to lower-fit participants while no differences were observed in a temporal orienting task and a duration perception task. Recently, the same research group (*Luque-Casado et al., 2016a*; *Luque-Casado et al., 2016b*) has extended their findings identifying potential neural and autonomic physiological mechanisms underlying the fitness-related improvements in vigilance performance in a 60′ version of the PVT. Their findings showed that a higher level of fitness was related to a better behavioural performance (i.e., in the first half of the task), that was accompanied by larger amplitude in the contingent negative variation (CNV) potential and enhanced cardiac orienting reflex. Crucially, high-fit individuals also maintained larger P3 amplitude throughout the task compared to low-fit, who showed a reduction in the P3 magnitude over time. The authors finally concluded that a higher level of fitness level was related to electrophysiological activity suggestive of better ability to allocate attentional resources over time and a greater attentive preparatory state.

The PVT used by Luque-Casado and collaborators (*2013*; *2016a*; *2016b*) is a sensitive and reliable sustained attention reaction time (RT) task that requires participants to react as fast as possible to the onset of a visual (or auditory) stimulus presented on every trial and that occurs at random inter-stimulus intervals. The task is characterized by the high

temporal uncertainty of stimuli and the low learning effects (*Dinges et al., 1997*; *Jewett & Kronauer, 1999*). The PVT has been commonly used to study the vigilance decrement by the repeated administration of the task in different moments of the experimental session (*Arnal et al., 2015*; *Van Dongen et al., 2003*).

Crucially, together with RT tasks like the PVT, sustained attention has been measured by means of accuracy (ACC) tasks, typically involving the presentation of infrequent and unexpected targets presented among frequent non-target stimuli for relatively long periods of time (*Bunce, 2001*; *Getzmann, Falkenstein & Gajewski, 2013*; *Yagi et al., 1999*; *Pontifex et al., 2015*). In fact, the first approximation to sustained attention study using this type of task was the Mackworth Clock test (*Mackworth, 1948*). In this task, participants watched a second by second clock that occasionally click forward two seconds instead of one. Participants were asked to press a button whenever they notice the rare "two-seconds" event. Since this first study, ACC oddball tasks have been widely used in cognitive science to study sustained attention (*Parasuraman & Mouloua, 1987*; *García-Larrea, Lukaszewicz & Mauguiére, 1992*). These unspeeded tasks reduce motor demands and increase the uncertainty of the target onset. In fact, together with temporal uncertainty, the low probability of target appearance has been shown to be one of the major factors to tax sustained attention (*Parasuraman & Mouloua, 1987*). It is important to note that this is in line with the traditional view of sustained attention as a high-order cognitive function that enables the accurate detection of infrequent and unexpected targets presented in a sequence of frequent non-targets (*Norman & Shallice, 1986*; *Stuss et al., 1995*).

Luque-Casado and collaborators (*2013*; *2016a*; *2016b*) findings were based on the results from a RT task (i.e., PVT) where participants have to respond to every stimulus. Thus, an important issue remains unsolved: the potential association between aerobic fitness and the ability to detect and respond to relevant non-frequent target stimuli ignoring frequent irrelevant non-target stimuli, which, as noted above, has been considered one of the critical features to tax sustained attention. That is, there is still the question of whether the superior performance of high-fit individuals with respect to low-fit participants observed in an RT sustained attention task like the PVT would be also revealed in an ACC oddball task.

To address the aforementioned shortcoming of previous research, here, we compared performance of two groups of young adults differentiated in terms of aerobic fitness (high-fit and low-fit) in the RT PVT and an ACC oddball task. The 27′ duration of the oddball task allowed us to investigate the vigilance decrement as a function of aerobic fitness. It is important to note that previous studies have found significant vigilance decrements with even shorter task durations. For example, Nuechterlein and collaborators (*1983*) examined sustained attention performance in an oddball paradigm lasting only 8 min, and found a marked decrement in sensitivity over time. More importantly, *Ariga & Lleras (2011)* found a significant vigilance decrement in young adults after the first 10′ of the same oddball task used in the present study. A 5′ PVT presented before and after the oddball task was used to investigate the changes in RT performance as a function of time and aerobic fitness, and to try and replicate Luque-Casado et al.'s (*2013*; *2016a*) previous results. A shorter (with respect to *Luque-Casado et al., 2013*) version of the PVT was used in order not overload to the experimental session, as the main focus of the study was
the oddball task. Note also that previous research has tested the reliability of this shorter version of the PVT (*Loh et al., 2004*; *Roach, Dawson & Lamond, 2006*).

We expected that high-fit young adults would be faster in the PVT with respect to the low-fit participants. Similarly, we hypothesized that low-fit young adults would exhibit a larger accuracy and RT impairment along the time in both tasks than their high-fit counterparts.

## MATERIALS & METHOD

### Participants

A minimum sample size of 22 participants per group was required for a power level of .80 as determined by an *a priori* power analysis based on data from a previous study (*Luque-Casado et al., 2013*). Thus, forty-four volunteer undergraduate students were recruited through announcements on billboards of the University campus at the University of Granada, Spain, to be part of this study. Twenty-two participants (11 females) [1] with a low level of aerobic fitness were allocated to the low-fit group and twenty-two (11 females) participants with a high level of aerobic fitness were allocated to the high-fit group. The participants in the high and low-fit group met the inclusion criteria of reporting at least 8 h of training per week or less than 2 h, respectively. The ventilator aerobic threshold (VAT) of the subjects was then determined by an incremental effort test in order to confirm their aerobic fitness characteristics (see below). All participants had normal or corrected to normal vision, reported no neurological, cardiovascular or musculoskeletal disorders and were taking no medication. All subjects gave written informed consent before the study and received 10 euros for their participation. This study was approved by the Ethics Committee on Human Research of the University of Granada, Spain (No. 689) and complies with the ethical standards laid down in the 1964 Declaration of Helsinki.

### Apparatus and materials

All participants were fitted with a Polar RS800 CX monitor (Polar Electro Öy, Kempele, Finland) to record their heart rate (HR) during the incremental exercise test. We used a ViaSprint 150 P cycle ergometer (Ergoline GmbH, Germany) to induce physical effort and to obtain power values, and a JAEGER Master Screen gas analyser (CareFusion GmbH, Germany) to provide a measure of gas exchange during the effort test. All tasks were completed using a 15′6-inch LCD HP laptop PC and the E-Prime software (Psychology Software Tools, Pittsburgh, PA, USA) was used for stimulus presentation and data collection. The center of the laptop screen was situated at 60 cm, using a chin cup, from the participants' head. The device used to collect responses was the PC keyboard.

### Procedure

The study procedure consisted of a single testing session of 90 min approximately. At the beginning, we performed an anthropometric evaluation of each participant (height, weight and body mass index [BMI]) in a small, dimly lit room. Then, participants completed, in this order, the PVT (5 min), a visual oddball task (27 min) and the PVT (5 min). All cognitive tasks were completed at rest with a break of 5 min before and after the oddball task. The tasks are detailed in the following sections.

[1] Note that a previous study showed that performance in the PVT did not depend on gender (*Ballester et al., 2015*). We therefore decided to include the same number of males and females only for the purpose of experimental control.

### VAT determination test

At the end of the session, all subjects performed an incremental cycle-ergometer test in order to determine their aerobic fitness. The incremental effort test started with a 3 min warm-up at 30 Watts (W), with the power output increasing 10 W every minute. Each participant set his preferred cadence (between 60–90 rev min$^{-1}$) during the warm-up period and was asked to maintain this cadence during the entire protocol. The test began at 60 W and was followed by an incremental protocol of 30 W every 3 min. Each step of the incremental protocol consisted of 2 min of stabilized load and 1 min of progressive load increase (5 W every 10 s). The oxygen uptake (VO2 ml min$^{-1}$ kg$^{-1}$), respiratory exchange ratio (RER; i.e., $CO_2$ production $O_2$ consumption$^{-1}$), relative power output (W Kg$^{-1}$) and heart rate (bpm) were continuously recorded throughout the test.

We used the ventilatory anaerobic threshold (VAT) as a reference to determine the fitness level of the participants. VAT is considered to be a sensitive measure for evaluating aerobic fitness and cardiorespiratory endurance performance (*Londeree, 1997*; *Wasserman, 1984*). It was defined as the VO2 at the time when RER exceeded the cut-off value of 1.0 (*Davis et al., 1976*; *Yeh et al., 1983*) and did not drop below that level during the 2 min constant load period or during the next load step, never reaching the 1.1 RER. The submaximal cardiorespiratory fitness test ended once the VAT was reached.

### Psychomotor vigilance task

A modified version of the PVT developed by *Wilkinson & Houghton (1982)* was used here. A black circle with a red edge (6.68° × 7.82°) was displayed at the center of the screen in a black background. The circumference began to be filled in a red color and in a counter-clockwise direction with an angular velocity of 0.094 degrees per second in a random time interval (from 2000 to 10000 ms) after the appearance of the black circle. After that, the participants had a total time limit of 1500 ms to respond. Participants were instructed to press the space bar with the index finger of their dominant hand as fast as possible when the circumference started to be filled, avoiding anticipation. Feedback of the response time was displayed on the screen on each trial for 300 ms. Data from trials with RTs below 100 ms (2.72%) were considered anticipations and therefore discarded from the analysis. Following the response to the target, or after 1500 ms in case of a missed response (if a response was not made during this time, the message "You did not answer" appeared on the screen), the next trial began. The task lasted for 5 min without interruptions.

### Oddball task

A visual oddball task based on that used by *Ariga & Lleras (2011)* was designed to measure sustained attention. The task consisted of a sequence of vertical, white and one-pixel thick lines displayed in the center of screen on a black background. A red circle was continuously visible at the center of the display as fixation point. The participants were instructed to respond to the target line (short line, 10.47°) pressing the "B" key with their index finger of their dominant hand and not to respond when the non-target line (long line, 11.99°) was shown. Participants were encouraged to respond accurately, not quickly. The task consisted of two blocks of 13.5 min, without any break between them, with the
target stimuli pseudo-randomly presented in 10% of trials (with the restriction that thirty targets appeared on each block) and non-target stimuli in 90%. The stimulus duration was 153 ms, and a variable interstimulus interval (1.3–2.3 s) was used. The task lasted for 27 min without breaks.

### Design statistical analyses

The participants' descriptive and fitness data were analyzed using $t$-test for independent samples.

For the PVT, a two-way analysis of variance (ANOVA) with the between-participants factor of aerobic fitness (high-fit, low-fit) and the within-participants factors of session (pre, post) was applied to analyze participants' mean RT.

For the oddball task, a two-way ANOVA with aerobic fitness (high-fit, low-fit) as between- subject factor and time-on-task (block 1, block 2) as within-subject factor was used to analyze (separately) participants' ACC (hits) and RTs to target trials, and ACC (commission errors) to non-target trials.

All analyses were performed with the SPSS statistical package. Multiple contrasts using $t$ tests with Bonferroni adjustment were performed when appropriate. All alpha levels for significance were set as .05 prior to Bonferroni adjustment. Corrected probability values are reported. Effect sizes of significant results in the ANOVA are reported as partial eta-squared ($\eta p^2$) for Fs and as Cohen's d for $t$-tests.

## RESULTS

### Descriptive and fitness data

The $t$-tests for independent samples revealed significant differences between groups in all the incremental test parameters (i.e., maximum power output (W), $t(42) = 5.437$, $p < .001$, $d = 1.67$, relative power output (W kg$^{-1}$), $t(42) = 5.488$, $p < .001$, $d = 1.69$, VO$_2$ (mL min$^{-1}$ kg$^{-1}$), $t(42) = 4.595$, $p < .001$, $d = 1.41$, and HR (bpm), $t(42) = -3.987$, $p < .001$), $d = -1.25$. Data showed clear evidence of the difference in aerobic fitness between groups (see Table 1). There were no statistically significant differences between groups in any other descriptive data (all $ps \geq .5$).

### PVT

The ANOVA of participants' mean RTs revealed a marginally significant main effect of aerobic fitness, $F(1, 42) = 3.663$, $p = .052$, $\eta p^2 = .08$. The high-fit group responded faster overall than the low-fit group (323 ± 24 ms and 346 ± 46 ms, respectively). Neither the main effect of session, $F < 1$, nor the interaction between both factors reached statistical significance, $F < 1$.

### Oddball task

*Hits trials.* The analysis of ACC (Hits) for target-trials yielded a significant main effect of time-on-task, $F(1, 42) = 14.207$, $p < .001$, $\eta p^2 = .25$, that was better qualified by the significant interaction between aerobic fitness (high-fit, low-fit) and time-on-task (block 1, block 2), $F(1, 42) = 4.502$, $p = .040$, $\eta p^2 = .09$; see Fig. 1). The $t$-test for dependent samples between block 1 and block 2 reached significance for the low-fit group, $t(21) = 4.069$,

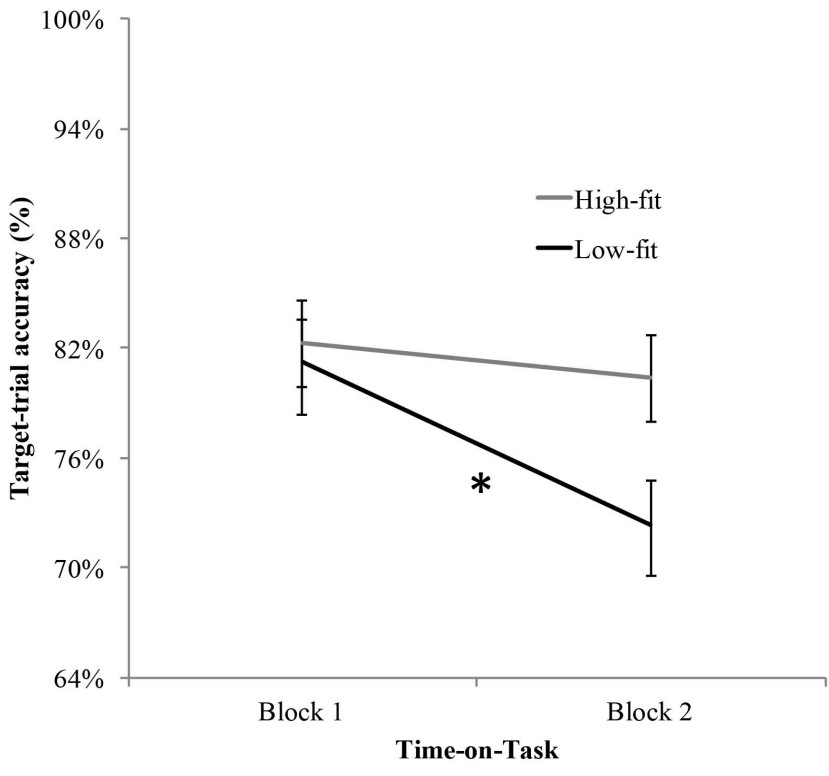

**Figure 1** **Mean response ACC (%) and standard error for target-trials as a function of Group and Block.** * Significant effect of time-on-task.

**Table 1** **Mean and standard deviation of descriptive and fitness data for the high-fit and low-fit groups.**

|  | High-fit | Low-fit |
| --- | --- | --- |
| **Anthropometrical characteristics** |  |  |
| Sample size ($n$) | 22 (11 females) | 22 (11 females) |
| Age (years) | 22.3 ± 3.5 | 22.5 ± 3.3 |
| Height (cm) | 170 ± 7.8 | 168 ± 10.8 |
| Weight (kg) | 66.2 ± 9.9 | 67.2 ± 16.8 |
| BMI (kg/m$^2$) | 22.7 ± 2.7 | 23.3 ± 3.9 |
| **Incremental test parameters** |  |  |
| Power max at VAT (W) | 195 ± 70.4 | 103.8 ± 34.9 |
| Relative power at VAT (W/kg) | 2.9 ± 1.1 | 1.6 ± 0.5 |
| VO$_2$ at VAT (ml/kg/min) | 35.9 ± 13.9 | 20.8 ± 6.3 |
| Heart rate (bpm) | 61.7 ± 12.7 | 79.2 ± 15.8 |

$p < .001$, $d = 0.48$, showing a reduced ACC over time ($81.2 \pm 17$ % and $72.4 \pm 18$ %, in block 1 and block 2, respectively). In contrast, the difference between blocks was not significant for the high-fit group, $t(21) = 1.194$, $p = .246$, $d = 0.17$), ($82.3 \pm 12$ % and $80.1 \pm 15$ %, in block 1 and block 2, respectively). The analysis of RTs for target-trials yielded no significant main effects of time-on-task, $F < 1$, and aerobic fitness, $F(1, 42) = 2.855$, $p = .099$, $\eta p^2 = .06$. The interaction between both factors was not significant either, $F < 1$.

*Commission errors.* The maintenance of accuracy in target-trials during the task showed by high-fit participants could have been due to a mere higher response rate than low-fit participants. However, this would have resulted also in more commission errors (responses to non-targets). To address this issue, we analyzed participants' mean ACC (commission errors) to non-targets. The ANOVA showed a main effect of time-on-task, $F(1, 42) = 7.579$, $p = .009$, $\eta p^2 = .15$, with participants being more accurate in block 2 ($96.3 \pm 12$% and $92.3 \pm 11$%, for high-fit and low-fit, respectively) than in block 1 ($95.1 \pm 12$% and $88.8 \pm 15$%, for high-fit and low-fit, respectively). The main effect of aerobic fitness failed to reach statistical significance, $F(1, 42) = 3.680$, $p = .062$, $\eta p^2 = .08$. There was no significant interaction between both factors $F(1, 42) = 1.549$, $p = .220$, $\eta p^2 = .03$.

## DISCUSSION

In the present study, we investigated the role of aerobic fitness in the sustained attention capacity of young adults. Towards this aim, two groups of participants (high-fit and low-fit) completed the PVT and an oddball task.

The results of the PVT showed that high-fit young adults were faster than low-fit, replicating previous research (*Luque-Casado et al., 2013*; *Luque-Casado et al., 2016a*; *Luque-Casado et al., 2016b*). Interestingly, even though the statistical analysis showed a marginal ($p = .053$) effect, the 23 ms difference between high-fit and low-fit participants was larger than the 19 ms reported in *Luque-Casado et al. (2013)* and the 12 ms (in the first 10′ of the task) reported in *Luque-Casado et al. (2016a)*. Taken together, the present and previous research from our laboratory supports the PVT as a suitable tool to measure group differences in terms of RT performance in a vigilance task involving high temporal uncertainty of target onset. Note also, that previous studies (e.g., *Dorrian, Rogers & Dinges, 2005*; *Drummond et al., 2005*; *Lim & Dinges, 2008*) showed the recruitment of the sustained attention brain network during the PVT, highlighting the sensitivity of the task to vigilance demands, over and above the time-on-task effect. Indeed, *Drummond et al. (2005)* found different activation patterns of cortical and subcortical brain regions during the fastest RTs of a PVT, relative to the slowest RTs. They suggested that the activity of these regions might reflect top-down modulation processes related to the capacity to maintain the focus on the task. In the present study, high-fit and low-fit participants appear to differ in their overall RT performance, but not in the effect of session. The lack of a session effect in the PVT might be due to the time break before the second PVT and even because of the presence of the oddball task between PVT sessions, which might have provoked a sort of "mind reset" (cf. *Ariga & Lleras, 2011*), preventing vigilance degradation with respect to the first PVT.

Nevertheless, it is also possible that the PVT employed in the current manuscript was not demanding sustained attention enough to induce the RT increment from the first to the second PVT session.

Importantly, in the oddball task, the high-fit group maintained their ACC rate over the whole duration of the task, while the low-fit group suffered a significant decline of response ACC over time. This finding suggests that aerobic fitness is related to the capacity to focus on relevant infrequent stimuli ignoring irrelevant frequent stimuli. Interestingly, the lack of group differences in RT in the oddball task seems to indicate that the greater sustained attention capacity showed by high-fit participants was irrespective of the nature of the task, i.e., high-fit young adults seem to adapt their performance to the goals of each task, either speed or ACC. Note, though, that the task instruction (stressing ACC over RT) in the oddball task might also explain the absence of RT difference between groups.

The present findings could be interpreted in the context of the "cardiovascular hypothesis" which proposes that exercise, particularly aerobic exercise, results in measurable gains in cardiorespiratory fitness, leads to benefits at the cognitive and brain levels. These gains refer to an increased ability of the heart to deliver oxygen to the working muscles which is thought to be associated with changes in cerebral structure (*Colcombe & Kramer, 2003*), cerebral blood flow (*Endres et al., 2003*), and a brain-derived neurotrophic factor [BDNF, IGF-1] (*Vaynman, Ying & Gomez-Pinilla, 2003*). In turn, all these changes have themselves been shown to be associated with cognitive performance (*Vaynman, Ying & Gomez-Pinilla, 2004*; *Hillman, Erickson & Kramer, 2008*; *Voss et al., 2013*; *McMorris, 2016*). Thus, according to the cardiovascular fitness hypothesis, changes in aerobic fitness are necessary for the cognitive benefits of physical activity to be observed. Certainly, the higher levels of aerobic fitness of high-fit young adults in our study may be associated with changes in underlying physiological mechanisms that result in a greater capacity to maintain attention over time compared to low-fit young adults. However, it is important to note that several meta-analyses of available studies seriously question this hypothesis (*Angevaren et al., 2008*; *Verburgh et al., 2014*; *Young et al., 2015*). This means that the exact mechanisms that produce the observed enhanced cognitive performance in physically fit individuals are still unknown.

At this point, it is also important to note that the purpose of this study was not to establish a causal-effect relationship, but rather to investigate the potential association between sustained attention and aerobic fitness. Randomized controlled trials are needed to establish that cause–effect relationship with respect to sustained attention and aerobic fitness. Moreover, we believe that future studies should investigate complex organ interactions (e.g., brain-heart communication) as a mechanism contributing to the enhanced cognitive and brain function in physically fit individuals (see *Perakakis et al., 2017*, for a first approach to this issue).

## CONCLUSIONS

To conclude, the current research contributes to further understand the positive relation between aerobic fitness and sustained attention capacity in young adults. Importantly,

a profound knowledge of the factors that support and enhance sustained attention is especially critical for public health, as a sedentary lifestyle has been blamed for so many diseases and also deterioration in cognitive function and performance of many daily life activities.

### Funding

This research was supported by research grants from the Ministerio de Economía y Competitividad (PSI2013-46385-P) and the Junta de Andalucía (SEJ-6414) to Daniel Sanabria. The funders had no role in study design, data collection and analysis, decision to publish, or preparation of the manuscript.

### Grant Disclosures

The following grant information was disclosed by the authors:
Ministerio de Economía y Competitividad: PSI2013-46385-P.
Junta de Andalucía: SEJ-6414.

### Competing Interests

The authors declare there are no competing interests.

### Author Contributions

- Luis F. Ciria conceived and designed the experiments, performed the experiments, analyzed the data, contributed reagents/materials/analysis tools, wrote the paper, prepared figures and/or tables, reviewed drafts of the paper.
- Pandelis Perakakis and Daniel Sanabria conceived and designed the experiments, contributed reagents/materials/analysis tools, wrote the paper, reviewed drafts of the paper.
- Antonio Luque-Casado conceived and designed the experiments, contributed reagents/materials/analysis tools, reviewed drafts of the paper.
- Cristina Morato performed the experiments.

### Human Ethics

The following information was supplied relating to ethical approvals (i.e., approving body and any reference numbers):

This study was approved by the Ethics Committee on Human Research of the University of Granada, Spain (No. 689) and complies with the ethical standards laid down in the 1964 Declaration of Helsinki.

### Data Availability

Ciria, Luis (2017): SAAF2: Sustain Attention and Aerobic Fitness. figshare.
https://doi.org/10.6084/m9.figshare.4977815.v3

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
