# Peer review of "The relationship between sustained attention and aerobic fitness in a group of young adults"

_PeerJ, doi:10.7717/peerj.3831_

## Round 0.1 · original submission · Major Revisions

I now have received three reviewers' comments. Although all of them expressed their interest in your study, several aspects of this manuscript should be revised to improve its clarity. Their observations are presented with clarity so I'll not risk confusing matters by belaboring or reiterating their comments. While I might quibble with the occasional point, I note that I regard the reviewers' opinions as substantive and well-informed. I believe that all of the highlighted reservations require contemplation and appropriate attention in revising the document if it is to contribute appropriately to Peerj and the extant literature. Please revise or refute according to the reviewers' comments and provide a point by point reply in addition to the revised manuscript. In addition, reviewers also pointed out the language issue that dramatically impaired the quality of your manuscript. Therefore, I'd suggest you to have your revised manuscript gone through a thorough language editing by a professional native speaking editor before your resubmission.

Tsung-Min Hung, Ph.D.
PeerJ editor
Distinguished professor
Department of Physical Education
National Taiwan Normal University

·

Basic reporting

The present manuscript is relatively well write. However, several part need to be carefully reread to correct several inattention mistakes.
- Line 51: incongruent date and time
- Line 69-71: the sentence begin in english and finish in spanish
- Line 79: comments in spanish and annotation
References in the text should be check. There are several redundancy in name or bracket wrongly used line 73, 79, 317.

I am not relevant to judge the quality of English.

The working hypothesis could be detailed at the end of the introduction.

The table 1 is cited at page 7 but only insert in page 11.

Experimental design

The present study aimed to investigate the impact of aeobic fitness on sustained attention in young adults. A high-fit group and a low-fit group completed a 5' PVT task (two times, respectivelly before and after an oddball task) and a 27' oddball task.

Method:
Line 159: precise that the cognitve tasks were performed at rest.
Line 160: precise taht the incremental test was performed in order to to determine the aerobic fitness caracteristic of the subjects (i.e., VAT, VO2 at VAT, power at VAT).
The description of the VAT determination test should be presented in a separate paragraph like the 2 cognitive tasks, intitulated "VAT determination test" for exemple.
The sentence "The filled circumference was presented for 500 ms" make the description of the PVT confusing, I suggest to delete it or move it somewhere else in the paragraph (line 185 for example).
Line 198: Contrary to the PVT, you encourage your subjects to respond accurately, not quickly. Why? this could explained part of your results, especially the afct that you have no effect on RT in the oddball task. This need to be discuss.
Line 202: The authors indicate to late (line 208-209) that the oddball task contained 2 distinct blocks. This information shuld be given earlier (line 202). Did the subjects have a short break between the 2 blocks?

Validity of the findings

The results of the oddball task would be more clear if the results would be presented in two separated paragraphs. The first one concerning 1/ Hit trials and the second 2/the commission errors.
Line 238: precise that the interaction you are talking about is the fitness x TOT interaction.

Figure 1 and 2: please add symbols representing the significant effects on the graph. Change the title of the y-axes, respectively by "Traget trial accuracy" for the Fig 1 and "Non target trial accuracy" for the fig2.

Add the means in the text in the results section.

Line 256-258 should be move in the discussion section or deleted.

The results confirm that high-fit subjects performed better on the PVT task than low-fit subjects. However, no effect of TOT task and no interaction with aerobic fitness was observed. We could have expected a degradation of the perfomance over time (between pre and post PVT), at least for low-fit subjects. How do you explained that?

Line 295-297: reverse the sentence.

You should discuss the fact that the oddball task is perhaps not enough sensitive to highlight the effect of fitness on sustained attention. Moreover, the accuracy instructions given for this task in particular could have altered the results.

Line 315: change VO2 by VO2 at VAT.
Do you have any idea of a better predictor?

Additional comments

The present study is of interest, however some modifications are necessary before publication. I hope that this comments will be helpful.

Reviewer 2 ·

Basic reporting

This current study examined the relationship between sustained attention and aerobic fitness in young adults using oddball and PVT cognitive tasks in young adults. Results indicated that higher-fit adults maintained response accuracy over time, while low-fit adults suffered a decline of response accuracy throughout the task, suggesting that higher fit adults have a great sustained attention. This manuscript was well written and the results were significant.

Minor change:

1. Line 70-71: please rewrite the sentence as it was not written in English.
2. Line 155: "approx" is not a complete word.
3. Line 210: "using 1-way between-groups design" seems not clear regarding of the proper description for statistical analysis.

Experimental design

no comments

Validity of the findings

no comments.

Additional comments

no comments

Reviewer 3 ·

Basic reporting

This manuscript focuses on the relationship between aerobic fitness and sustained attention assessed by a PVT and an oddball task emphasizing response time and accuracy, respectively. While I believe that this manuscript is interesting, there are a number of potential issues that should be addressed prior to this manuscript being considered for publication.

 Please make sure the format of in-text citation and reference is consistent throughout the manuscript as well as in line with PeerJ’s guideline. For example, be consistent in terms of whether capitalizing the first letter of each word in article and journal title in the reference section. Also, check in-text citation such as line 79 and 317.
 Some texts in the manuscript were not written in English (line 70, 79..). Please proofread and modify accordingly.
 I would recommend the authors to change the title given that the current title reads like only oddball task was used.
 Line 81: the authors may need to explain what were the neuroelectric indices for sustained attention or response preparation in Luque-Casado et al. (2016).
 Line 309: these two citations regarding brain activation during exercise are odd at here. Instead, the authors should cite prior research examining the effects of fitness or chronic participation in physical activity on structural and function adaptations in the brain.

Minor issues
1. Some grammar errors need to be corrected such as line 6 in the abstract ("divided in" should be "divided into") and line 313 ("limit" should be "limits"). Please proofread or have a English native speaking colleague review the manuscript.
2. Line 46: I would recommend the authors avoid stating “any type of information processing” as there may be processes that require little of sustained attention such as some bottom-up or automated processing.
3. Line 51: there are irrelevant numbers at the end of the sentence.
4. Line 197: it may be helpful to add “non-target” before “long line”.

Experimental design

 What is the unique contribution of this study? Why is it important to examine the relationship between aerobic fitness and sustained attention using an oddball task emphasizing accuracy? In other words, does response accuracy during an oddball task reflect one aspect of sustained attention that is different from RT during PVT? The authors may need to discuss why investigating the effect of fitness on accuracy sustained attention task is necessary for filling the knowledge gap in the field.
 Line 124: It is not clear why the authors adopted two blocks of 5 min PVT task separated by a 27 min oddball task. If the purpose of this study was to replicate Lique-Casado et al., (2013, 2016), why not used a 10-min block or a 60-min block? Please justify.
 Line 131: please report the effect size that was powered to be detected based on the sample size in the current study.
 Line 134: Were participants allocated to low- and high-fit groups according to the frequency of weekly training? Does the training mean sports training or physical activity? And how to define training and physical activity in terms of intensity? The description (line 134-138) may need to be modified for better clarity.


Minor issues
1. Line 208: The authors may consider moving this sentence to the Oddball task section.

Validity of the findings

 Line 260: The significant fitness main effect on non-target trials does not support the selective improvement hypothesis discussed in line 302 given that high-fit participants outperformed their low-fit counterpart for both target and non-target trial types. The authors need to include trial type as a factor to test whether the beneficial effect of aerobic fitness on target trials is larger than that of non-target trials.
 The data from PVT did not show an impaired performance over time for both groups. Was PVT used in this study a valid measure of sustained attention?
 Line 321-322: Instead of their own belief, the authors should discuss what the existing literature found. For example, McMorris et al. (2016) and Voss et al. (2013) provided very comprehensive reviews regarding the possible mechanisms to account for the beneficial effects of chronic and acute physical activity on cognition.
 Line 289: the statement “this later findings supports the PVT as a suitable tool to measure group differences in terms of RT performance in a vigilance task involving high temporal uncertainty of target onset” is fine. However, such fitness main effect may be not due to modulation of sustained attention because RT was maintained over time in both low- and high-fit groups. The authors need to explain what is contributing to this fitness main effect on PVT performance and why it is not consistent with prior studies.
 It is not clear why the authors performed a correlational analysis among measures of aerobic fitness and sustained attention. Such analytical approach should generate similar results derived from factorial analyses in a cross-sectional study. It is strange that a fitness main effect was observed using factorial analyses but the correlation between fitness and measures of sustained attention is null. The related issues are below
I. The result of the correlational analysis is problematic because it suggests that aerobic fitness has no impact on sustained attention, which is contrary to the results derived from the factorial analysis. If the authors intended to include both of these analyses, they may need to explain such discrepancy.
II. It is not clear if aerobic fitness was correlated with the overall accuracy across block 1 and block 2 of the oddball task or with accuracy for each block separately.
III. Line 315-317: the two points based on the null correlation are confusing. In this study, VO2 reflected aerobic fitness. The null correlation only suggests that VO2 is not a good predictor for sustained attention. Similarly, the null correlation suggests that aerobic fitness, which is essentially VO2, cannot explain sustained attention capacities. Regardless, the correlational analysis performed in this study cannot generate any conclusion related to the so-called cardiovascular hypothesis.

---

## Round 0.2 · Minor Revisions

I have now received two reviewers’ comment and both reviewers were generally satisfied with your reply and revisions from previous comments. However, one reviewer has pointed out some issues that require your additional attention. Please address these issues and provide a point by point reply in addition to the revised manuscript.

Tsung-Min Hung, Ph.D.
PeerJ editor
Distinguished professor
Department of Physical Education
National Taiwan Normal University

·

Basic reporting

no comment

Experimental design

no comment

Validity of the findings

no comment

Additional comments

The authors have responded to all my comments. The manuscript has been greatly improved and deserves publication.

Reviewer 3 ·

Basic reporting

1.
Line 133: It may be helpful if the authors could cite studies that tested the reliability of short PVT in adults. If no such evidence exists, the authors may consider replacing “infants” by other terms that indicate children aged ~11 yr according to Wilson et al. (2010)

2.
To me, “mind reset” is a perfectly fine explanation for the lack of session effect in the PVT. However, another possibility is that the PVT (two 5’ blocks separated by oddball task) employed in the current manuscript was simply not demanding sustained attention/vigilance enough to induce the RT increment from the first to second block. If I understand correctly, line 284-287 might be intended to suggest that it is possible that PVT is a valid measure of sustained attention/vigilance even when time-on-task effect is not apparent or not observable. However, it may be helpful if the authors could explain more about this idea as well as the cited studies (e.g., Dorrian et al., 2005; Drummond et al., 2005; Lim & Dinges, 2008) with better clarity.

The authors then stated that “High-fit and low-fit participants appear to differ in their overall RT performance, but not in the effect of session. This would seem to nuance our statement of the superior vigilance capacities of high-fit participants.” I am not sure if the first sentence is based on prior studies or the current manuscript. If it’s referring to prior studies, please provide citations. If the sentence is based on the current study, it is redundant because the following sentence (Line 290-291) repeats the similar thing.

Thus, I would recommend revising the second half of this paragraph (line284-294) for better readability

3.
Please check the use of tense throughout the manuscript. For example, Line 101 Participants “were” asked; Line 227: Probability values “were” reported; Line 228: ANOVA “were” reported…

Experimental design

NA

Validity of the findings

NA

Additional comments

I commend for the authors' effort on addressing all of my concerns. This manuscript is substantially improved and can contribute to the literature. I believe this manuscript can be accepted for publication after the minor revisions listed above.

---

## Round 0.3 · accepted · Accept

I have read through your reply to the reviewer's comment and your revised manuscript. I am satisfied with your response and decided that there is no need to send to the reviewer. You and your coauthors have my congratulations. Thank you for choosing PeerJ as a venue for publishing your research work and I look forward to receiving more of your work in the future.

Tsung-Min Hung, Ph.D.
PeerJ editor
Distinguished professor
Department of Physical Education
National Taiwan Normal University